# Autistic traits foster effective curiosity-driven exploration

**Francesco Poli** [1,2] *, **Maran Koolen**[1], **Carlos A. Velázquez-Vargas**[3], **Jessica Ramos-Sanchez**[1], **Marlene Meyer**[1], **Rogier B. Mars**[1,4], **Nanda Rommelse**[5], **Sabine Hunnius**[1]

**1** Donders Institute for Brain, Cognition and Behaviour, Radboud University, Nijmegen, Netherlands, **2** MRC Cognition and Brain Sciences Unit, University of Cambridge, Cambridge, United Kingdom, **3** Princeton University, Princeton, New Jersey, United States of America, **4** Wellcome Centre for Integrative Neuroimaging, Centre for Functional MRI of the Brain (FMRIB), Nuffield Department of Clinical Neurosciences, John Radcliffe Hospital, University of Oxford, Oxford, United Kingdom, **5** Department of Developmental Psychology, Utrecht University, Utrecht, the Netherlands

* francesco.poli@mrc-cbu.cam.ac.uk

**Data Availability Statement:** Data and code for computational models and statistical analyses are available on OSF: https://osf.io/h2prm/ (DOI 10.17605/OSF.IO/H2PRM).

## Abstract

Curiosity-driven exploration involves actively engaging with the environment to learn from it. Here, we hypothesize that the cognitive mechanisms underlying exploratory behavior may differ across individuals depending on personal characteristics such as autistic traits. In turn, this variability might influence successful exploration. To investigate this, we collected self- and other-reports of autistic traits from university students, and tested them in an exploration task in which participants could learn the hiding patterns of multiple characters. Participants' prediction errors and learning progress (i.e., the decrease in prediction error) on the task were tracked with a hierarchical delta-rule model. Crucially, participants could freely decide when to disengage from a character and what to explore next. We examined whether autistic traits modulated the relation of prediction errors and learning progress with exploration. We found that participants with lower scores on other-reports of insistence-on-sameness and general autistic traits were less persistent, primarily relying on learning progress during the initial stages of exploration. Conversely, participants with higher scores were more persistent and relied on learning progress in later phases of exploration, resulting in better performance in the task. This research advances our understanding of the interplay between autistic traits and exploration drives, emphasizing the importance of individual traits in learning processes and highlighting the need for personalized learning approaches.

## Author summary

Research has long recognized that individuals display curiosity and explore their environments in order to learn. It is suggested that personal characteristics, including autistic traits, might influence how one engages in such exploratory behaviors. In this study, participants with varying levels of autistic traits participated in a game of locating hidden characters. We aimed to understand their decision-making process: which character they decided to engage with and for how long. Remarkably, participants with stronger autistic

**Funding:** This study was supported by a Donders Centre for Cognition internal grant to S.H. and R.B. M. ("Here's looking at you, kid." A model-based approach to interindividual differences in infants' looking behavior and their relationship with cognitive performance and IQ; award/start date: 15 March 2018), a VICI grant from the Netherland Organization for Scientific Research NWO to S.H. ("Loving to learn - How curiosity drives cognitive development in young children"; serial number: VI. C.191.022), a Wellcome Trust center grant to benefit of R.B.M. ("Wellcome Centre for Integrative Neuroimaging"; serial number: 203139/Z/16/Z), a EPA Cephalosporin Fund and Biotechnology and Biological Sciences Research Council to R.B.M. (BB/N019814/1). The funders had no role in study design, data collection and analysis, decision to publish, or preparation of the manuscript.

**Competing interests:** The authors have declared that no competing interests exist.

traits exhibited distinct exploration patterns, and in scenarios requiring persistence, their approach was particularly effective. This research underscores the importance of recognizing that individuals, especially those with autistic traits, may possess unique strategies for exploration and learning. This realization can guide educators and policy-makers in crafting more tailored learning environments. Furthermore, it emphasizes that the presence of autistic traits can be associated with specific strengths, reshaping our understanding and appreciation of neurodiversity.

## 1. Introduction

How agents define their own learning curriculum, actively selecting what they want to explore, plays a pivotal role in learning [1]. This aspect of exploration stems from an intrinsic drive to learn, rather than from maximizing any extrinsic rewards. For this reason, it is often called curiosity-driven [2–3]. Recent evidence shows that humans explore different environments and engage in different activities taking into account the learning opportunities they offer [4–6]. Specifically, it was shown that participants playing a learning game were more likely to stop exploring an environment when they were making only little learning progress. Moreover, they were more likely to select activities in which they expected to make more learning progress [4].

The relationship between individual differences in curiosity-driven exploration and aspects of personality and real-world behaviors remains largely unexplored. Recent research has shown that exploration levels are highly variable across individuals [7], and possibly related to personality traits such as impulsivity and risk-taking [7,8]. This provides initial evidence that the mechanisms underlying exploratory behavior may be influenced by individual differences in personality traits. In this perspective, the exact mechanisms that drive exploratory behavior may be confounded at the group level [9], and can only be properly identified and understood by accounting for individual differences in personality traits. To explore this, we tested how the cognitive mechanisms underlying curiosity-driven exploration related to autistic traits, which represent a significant source of variability in personality traits within the general population [10,11].

Although the sensory [12–14], cognitive [15–17], social [18,19] and communicative aspects [20–22] of autistic traits have been studied thoroughly, recent theoretical work and empirical findings have sparked interest in how autistic traits relate to individual differences in learning mechanisms [23,24]. It was found that stronger autistic traits were linked to less efficient learning about probabilistically aberrant events [25,26], and the learning of participants with high insistence on sameness was less robust to noise [27]. However, autistic and non-autistic participants performed in the same way in visual search and decision-making tasks [28,29], and when environments were volatile [30], and some studies also showed enhanced statistical learning abilities in autistic people [31]. None of these studies focuses on the active aspect of learning, which has received surprisingly little attention in relation to autistic traits.

A recent study [32] investigated how, during exploration, autistic traits relate to differences in the tolerance to prediction errors (i.e., the mismatch between expected outcome and actual outcome). Participants could freely move between two environments, while they had to discover their latent structure. Participants with fewer autistic traits had a greater tolerance for prediction errors before opting to abandon the environment. These findings are consistent with the fact that autistic traits have been linked to insistence on sameness, intolerance to errors, and difficulty dealing with sudden changes [33–35]. Yet, it remains unknown how

these traits might alter the balance between different curiosity drives, such as seeking out novelty [36], minimizing prediction errors [37], and maximizing learning progress [38].

The lack of tolerance for uncertainty may influence curiosity-driven exploration behavior in two different ways. The first possibility is that individuals with higher autistic traits scores may be more motivated to reduce uncertainty. Given that learning progress allows uncertainty to reduce, they might thus give more importance to learning progress, even when it comes in small amounts. This predicts a stronger relationship between learning progress and exploration for people with higher scores on autistic traits. A second possibility is that the intolerance to uncertainty may lead to avoiding uncertain situations as much as possible [35,39]. As a consequence, people who score higher on autistic traits might actively avoid prediction errors. From this follows that their exploratory decisions would be guided by the avoidance of prediction errors rather than an interest in learning progress.

To assess these hypotheses, we tested participants on a learning task in which they could freely interact with different animal characters on a screen. The animals appeared following probabilistic patterns and participants were instructed to predict the next location of the animal. We generated these hiding patterns so that participants could learn about the likely next hiding location of the animal and improve their prediction performance (i.e., make learning progress). We manipulated various sources of uncertainty to generate learnable situations that allowed measuring participants' predictions and learning. Crucially, we monitored when participants decided to stop exploring an animal's hiding pattern, which animal they picked next, and whether this depended on their learning progress, their prediction errors, or the novelty of the animals. To better assess their performance and exploratory drives, we fit participants' behavioral responses to a hierarchical delta-rule model [4]. This allowed us to obtain trial-by-trial estimates of each participant's prediction errors and learning progress on the task, as well as their expectations of how much learning progress and prediction error they might experience in future trials.

We examined how exploration drives are affected by autistic traits through relating trial-by-trial estimates of (expected) learning progress and prediction error to exploration choices and self- and other-reports of autistic traits. Given that the need for predictability might be the reason why some people entertain repetitive behaviors and have specialized interests, the primary focus of our analyses was on the subscale of insistence on sameness, which assesses these aspects [40,41]. However, the same predictions may hold for multiple subscales, or for the total scale for autistic traits, because multiple traits may influence exploration decisions in similar ways. Therefore, we also explored the possibility of general effects of autistic traits on exploration behavior, in addition to examining the subscale of insistence on sameness. By investigating these possibilities, we aim to gain a better understanding of how autistic traits affect curiosity-driven exploration and how individuals with higher autistic traits scores might differ in their exploration drives from individuals with lower scores.

## 2. Methods

### 2.1. Ethics statement

The study was approved by Radboud University's ethics committee (Approval number ECSW2016-0905-396). A digital written consent was obtained from each participant before starting the study.

### 2.2. Participants

Seventy-seven participants were recruited for this study. All participants were or had recently been university students. Seven participants were excluded from the analysis, as they indicated

that they did not understand the task. As a result, seventy participants were included in the final analysis. This final sample consisted of 14 men, 51 women, and 5 nonbinary persons aged 17 to 35 years ($M$ = 22.2, $SD$ = 4.2). We did not investigate whether participants had an ASD diagnosis and/or identified themselves as neurodiverse.

## 2.3. Procedure

Participants were tested online via Zoom (zoom.us). The task was coded in PsychoPy [42] and then uploaded to Pavlovia (pavlovia.org). Participants performed the task while on Zoom. Afterwards, they filled in the self-report version of the Adult Social Behavior Questionnaire (ASBQ) [43] and a set of additional questions in which they reported their level of involvement with the task. Lastly, the other-report version of the ASBQ was provided to participants who had indicated to be both willing and able to ask a parental figure to fill in a questionnaire.

## 2.4. Task

Participants were presented with four cartoon animals and were asked to pick one of them by clicking on it (Fig 1A). The selected animal would appear in the middle of the screen (Fig 1B). When participants clicked on the animal again, it would disappear, as it was "hiding" somewhere behind a row of rocks (setting 1), waves (setting 2), or a hedge (setting 3). Participants had to click on the hiding area to indicate where they expected the animal to reappear (Fig 1C). Upon button click, the animal reappeared revealing its true location (Fig 1D). The other three animals would remain visible on the screen at all times. At any moment, participants

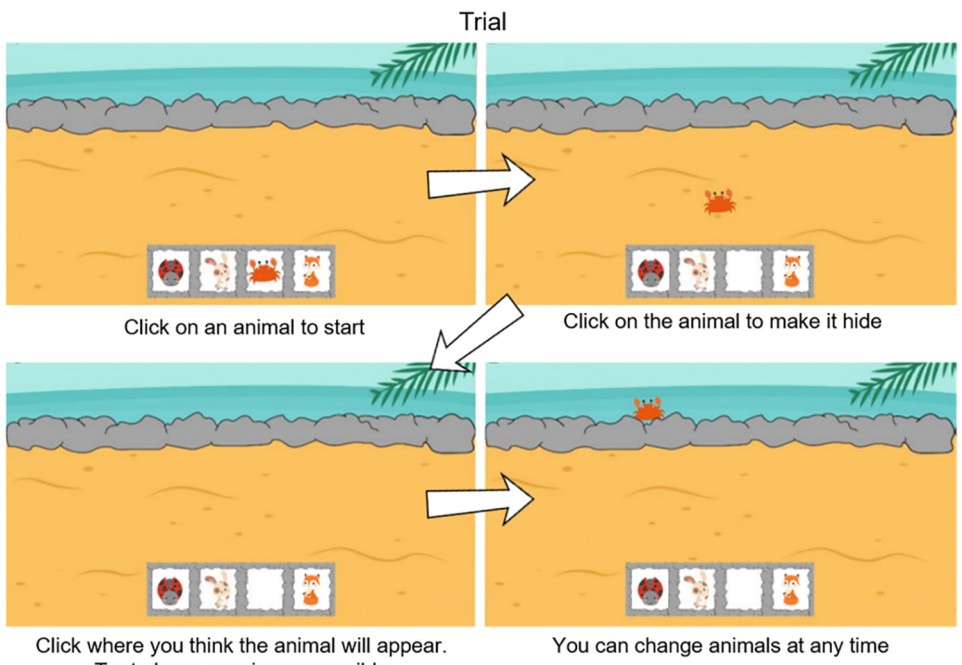

**Fig 1. Example trial from the learning task.** At the start of the task, participants could choose what animal they wanted to engage with. Then, the chosen animal appeared in the center of the screen, and the participant had to click on it to make it disappear. At this point, the participant had to indicate where they thought the animal was by clicking on that location on the screen. Finally, the correct position of the animal was revealed. Participants could switch between animals at any time, and also go back to animals that had been explored before. Note that the instructions were only available during the first trial that participants played. [Images were freely available on pixabay.com].

were free to decide whether to keep engaging with the same animal or switch to a different one. After any of the animals had hidden 35 times or the participant had played 90 times, the game advanced to the next setting. Participants were presented with three different settings in total (beach, sea, and grassland). In each setting, the participants could choose between four different animals to play with. After the third setting, the game ended. As a result, the overall number of trials varied between participants (range: 150–271). Participants received no instructions or visual cues about where the animals would hide or how to find them, and they received no external rewards for guessing the hiding spot correctly.

Each animal displayed a specific hiding pattern (see Fig 2A for examples): It appeared around a certain location (i.e., the mean position) with some degree of noise (i.e., the standard deviation of their position). Moreover, their position could slowly drift in a certain direction [44] or abruptly change [45]. By manipulating these parameters, four different patterns were created (Fig 2A). One pattern had low noise, low change-point probability, and a small drift. Another pattern had high noise, no change points and no drift. The third pattern had high change-point probability, low noise and no drift. The last pattern contained both high drift and high change-point probability, with low noise. These patterns were independent from each other, such that while the participant was playing with one of the animals, the locations of the other animals remained unchanged. Each setting featured a version of each of these four patterns. The specific animal displaying each pattern was counterbalanced across participants.

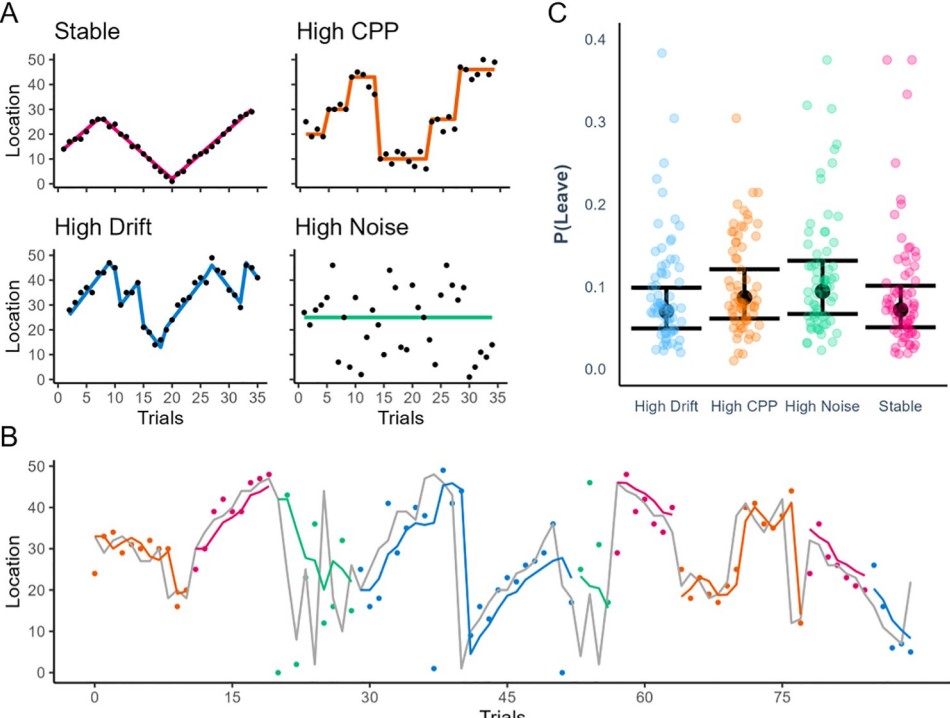

**Fig 2. A.** The hiding patterns were generated from Gaussian distributions, varying their drift, their change-point probability (CPP), and their standard deviation (i.e., noise). This resulted in four different patterns, three of which were learnable (i.e., all but the high noise pattern). **B.** Example data from one participant. Participants guessed the hiding location of the animals (coloured dots) and received feedback about their actual location (grey line). Changes in dots colour indicate that the participant switched from one animal to another. The delta-rule model inferred the beliefs of the participants (coloured lines) from their predictions (the dots). Learning progress, prediction errors, and novelty were computed for all patterns in the same way. C. Probability of disengaging from each pattern type. Dots indicates the average probability of each participant (see also Appendix B in S1 Text).

It should be noted that these patterns were introduced to generate variability in the task, and hence in the prediction errors and learning progress that participants would make. However, the patterns themselves were not the main focus of our analyses, but rather we were interested in *when* and *what* participants decided to explore.

We extracted three key variables from this task. First, participants' predictions about where the animal character would appear were extracted for each trial. This variable is the observable state that is used to fit the delta-rule model (see Fig 2B and the Methods section). Second, for each trial we measured whether participants chose a new animal or stayed with the same one. These are called leave-stay decisions (see Fig 2C). Finally, every time a leave decision was made and a new animal was chosen, we measured which animal was chosen among the three available options. These were called exploratory decisions. Leave-stay decisions and exploratory decisions were the main dependent variables of our paper, as we sought to understand when participants decided to explore, and what they decided to explore.

## 2.5. Questionnaire

After the task, participants filled in the Adult Social Behavior Questionnaire (ASBQ; Horwitz et al., 2016). The ASBQ is a multidimensional ASD Questionnaire, which has both a self-report and an other-report version. It consists of 44 items which are divided into six domains of ASD. These domains are reduced contact, reduced empathy, reduced interpersonal insight, violations of social conventions, insistence on sameness, and sensory stimulation and motor stereotypies. Using a 3-point Likert scale, participants indicate how strongly they feel a given statement applies to them. Item scores are summed to generate both, a total score as well as subscale scores for each domain. All participants returned the self-report questionnaire, and 60% of the participants (42/70) returned also the other-report questionnaire.

In the analyses, we primarily focused on other-reports collected from a parental figure. This decision was made based on the fact that individuals who have stronger autistic traits may also have more difficulty reflecting and reporting on their behavioral and emotional patterns or they may mask their patterns, which may result in more unreliable self-reports [46]. Indeed, previous research has found discrepancies between self- and other-reports [47]. For this reason, we expected parental reports to be more reliable. However, we always present results both for parental-reports and self-reports.

After filling in the ASBQ questionnaire, participants were asked a set of additional questions about the task, regarding when they got bored (first, second, third setting, or never), which animals were their favorites and least favorites to play with, and their subjective experience of the task (as 5-point Likert-scale statements, see Appendix A in S1 Text). Participants who indicated that they did not understand the task instructions were excluded from the analyses (N = 7). From the included participants, we discarded the data from the settings following the moment they reported boredom (after the 1st setting for 21 participants, after the 2nd setting for 23 participants, and after the 3rd setting for 26 participants). These data were discarded before running any analysis, including the hierarchical delta-rule model. This decision was motivated by the fact that the focus of this research was intrinsically motivated exploration, which cannot be measured in conditions of boredom.

## 2.6. Computational modelling

We developed a Bayesian hierarchical model with a latent structure that can adapt to both abrupt and gradual changes in the environment. This allowed us to fit participants' learning of both gradual and abrupt changes in the hiding patterns, thus improving our fit to the data. In turn, this led to more refined estimates of (expected) prediction error and learning progress.

The core structure of the model consists of a standard delta-rule algorithm that incorporates two types of learning. To learn about gradual changes, the model is endowed with an additional delta-rule algorithm that tracks the drift in the movement of the animals, rather than their position. To learn in the presence of abrupt changes, the model can quickly increase its learning rate, so that the incoming prediction error receives greater importance.

The learning of gradual changes was implemented with a simple delta-rule:

$$V'_{t+1} = V'_t + \beta(r_t - V_t) \tag{1}$$

where $V'_{t+1}$ is the predicted drift of the animal character for trial $t + 1$, which is updated using prediction errors weighted by the learning rate for drift, $\beta$. The prediction errors are the difference between the real location of the animal $r_t$ and the previous prediction about the animal location $V_t$.

In addition to gradual changes, the model could learn to adapt to abrupt changes by flexibly adjusting its learning rate $\alpha$:

$$V_{t+1} = V_t + V'_t + \alpha^*_t(r_t - V_t) \tag{2}$$

where $V_{t+1}$ is the predicted position of the animal for the $t + 1$ trial and is updated with prediction errors weighted by the learning rate $\alpha$. Crucially, the learning rate differs from classical delta-rule models, because it is composed by two $\alpha$ parameters instead of one. One $\alpha$ parameter assumes lower values than the other, thus leading to better learning when the environment is stable, while the other $\alpha$ parameter assumes higher values, thus learning to quickly adjust when abrupt changes occur. Hence, the learning rate $\alpha^*_t$ is decided independently for each trial (see the inner plate on Fig 3), depending of the needs of the moment. If a slow update is needed, the lower $\alpha$ is more likely to be chosen. If a fast update is needed, the higher $\alpha$ is more likely to be chosen. Whether the higher or lower alpha is employed for a specific trial is regulated by a binary indicator $z_t$ controlled by a Bernoulli distribution with parameter $\phi$. If the indicator equals 1, the higher learning rate is chosen; if it equals 0, the lower learning rate is used. Different parameters $\alpha$, $\phi$, and $\beta$ were fitted for each animal (see the most external plate in Fig 3), and were the same across all individuals.

Given this learning, the model could then make a guess $B$ about the animal's location:

$$B_{i+1} \sim N\left(V_{i+1}, \frac{1}{\eta}\right) \tag{3}$$

where the model's guesses (or responses) about the animal's location are assumed to be sampled from a Gaussian distribution with mean $V_{i+1}$ and variance $\eta$, representing the noise in the decision process. The noise $\eta$ was also fitted independently for each animal, and it was assumed to be shared across individuals. A Bayesian graphical representation [48] of this model is shown in Fig 3.

Five variables of interest were computed from the model to predict participants' exploratory behavior. The first variable of interest was the unsigned prediction error that participants made on each trial:

$$PE_t = |r_t - V_t| \tag{4}$$

From the trial-by-trial difference in prediction errors, it was possible to compute the learning progress:

$$LP_t = PE_{t-1} - PE_t \tag{5}$$

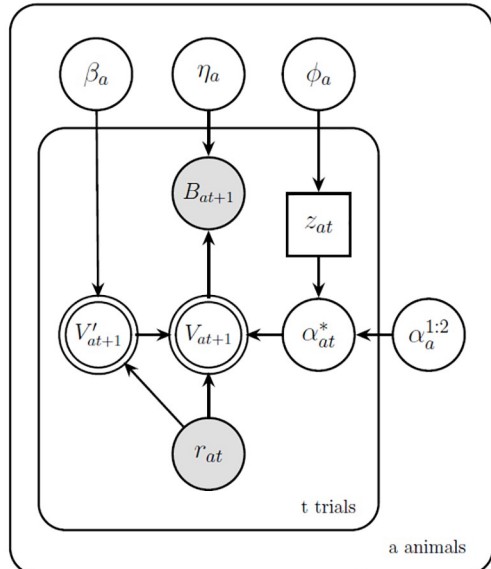

$$\phi_a \sim \mathrm{Uniform}(0,1)$$

$$\beta_a \sim \mathrm{Uniform}(0,1)$$

$$\eta_a \sim \mathrm{Gaussian}(0, 0.001)_{T(0,\infty)}$$

$$z_{at} \sim \mathrm{Bernoulli}(\phi)$$

$$\alpha_a^1 \sim \mathrm{Uniform}(0,1)$$

$$\alpha_a^2 \sim \mathrm{Uniform}(0,1)$$

$$\alpha_a^{1:2} = \{\alpha_a^1, \alpha_a^2\} : \alpha_a^1 \le \alpha_a^2$$

$$\alpha_{at}^* = \begin{cases} \alpha_a^1 & if \quad z_{at} = 0 \\ \alpha_a^2 & if \quad z_{at} = 1 \end{cases}$$

$$V_{at+1}' = V_{at}' + \beta_a(r_{at} - V_{at})$$

$$V_{at+1} = V_{at} + V_{at+1}' + \alpha_{at}^*(r_{at} - V_{at})$$

$$B_{at+1} \sim \mathrm{Gaussian}(V_{at+1}, \tfrac{1}{\eta_a})$$

**Fig 3. The hierarchical delta-rule model.** Nodes represent variables in the model and arrows connecting them their dependencies. Shaded nodes correspond to observed variables and unshaded nodes to latent variables. Stochastic and deterministic variables are represented using single- and double-boarded nodes, respectively. Continuous variables are represented using circular nodes and discrete variables with squared nodes. Finally, rectangular plates enclosing the nodes indicate replications of the process inside them. On the right-hand side of the figure, the assumptions about the parameters before data is observed and the learning equations are shown. The predictions made by the participants (B) are fitted as a gaussian distribution with a mean $V$ and precision $1/\eta$, where V is the expected mean location of any given animal, and $1/\eta$ captures the amount of certainty about the animal's location. $V$ is learned with a delta-rule, where the previous expectation about $V$ is updated by integrating velocity $V'$ and the prediction error weighted by the learning rate $\alpha$. Velocity is itself learned via a separate delta-rule, where the previous expectations about velocity are updated with the prediction error weighted by a learning rate $\beta$. The learning rate $\alpha$ can take two values, one greater than the other, which is determined specifically for each trial $t$ of each animal's pattern $a$ (see plating). The choice of $\alpha$ depends on a binary indicator $z_t$, which is determined probabilistically by the parameter $\phi$. All parameters were estimated across all individuals, but separately for each animal $a$. See Fig E in S1 Text for model comparison.

The learning progress is signed, as an increase in prediction error over time implies a worsening of the performance, while a decrease in prediction error over time corresponds to an improvement in the performance. The novelty of each option at any given trial was computed as an inverse function of exposure to that option, hence:

$$N_t = -t \tag{6}$$

where $t$ is the number of trials played with the same animal character. Two additional variables of interest were computed following previous modelling work [4], the expected prediction error and the expected learning progress. These variables capture how much prediction error and how much learning progress participants *expect* to make in the following trial. The expected prediction error is computed as follows:

$$PE_{expected_{t+1}} = PE_{expected_t} + \alpha_t^*\left(PE_t - PE_{expected_t}\right) \tag{7}$$

This is a delta-rule equation that mirrors the ones given above and uses the same learning rate $\alpha_t^*$. However, instead of updating the expected location or drift of the animal, it updates the expected prediction error for the next trial. As the learning progress is computed from the difference in prediction error between consecutive trials, the expected learning progress can be

computed from the difference between the expected prediction error and the actual prediction error:

$$LP_{expected_{t+1}} = PE_t - PE_{expected_{t+1}} \qquad (8)$$

A model was fit for each animal of each setting independently. Hence, at every trial, only the model of the animal that is selected is updated, while the others remain the same. To improve the stability of the trial-by-trial estimates, the parameters were fitted across participants. The models were fitted in JAGS, using three chains with 20,000 samples each, where the first half of samples were discarded as burn-in. All parameters reached convergence, as indexed by $\hat{R}$ values below 1.05. Details about performance simulation, parameters recovery, and model comparison can be found in the supporting information (Figs A-E in S1 Text).

## 3. Results

### 3.1. Influence of autistic traits on stay-leave decisions

We compared the goodness of fit of four logistic models to predict leave-stay behavior (Table 1). The models included a subset of the following variables: Prediction error, learning progress, novelty, expected prediction error, and expected learning progress. Current and expected estimates of prediction error or learning progress were not included in the same models. In every model, self- or other-reports of autistic traits and their interaction with the other variables were included to study their effect on leave-stay decisions. Autistic traits were treated here as a continuous variable. Finally, following previous research, the effect of time and its interaction with all other variables was added (cf. [4]), where time is computed as the number of consecutive trials played with a specific animal. Hence, all models had the following structure:

$$P(Leave_t) = logit\left(\beta_0 + \beta_{1:3}\begin{bmatrix} Novelty_t \\ (expected)\ PE_t \\ (expected)\ LP_t \end{bmatrix} * \beta_4 time * \beta_5 [Autistic\ Trait]\right)$$

The autistic trait of primary interest was insistence on sameness, but similar models were tested also with the full-scale of autistic traits and for each of the other subscales (reduced contact, reduced empathy, reduced interpersonal insight, violations of social conventions, sensory stimulation and motor stereotypies), either from other-reports or self-reports. Since other-reports are often more accurate than self-reports [49], the best-fitting model was obtained based on other-reports, and the same models were then run also with self-reports. All models included random effects for participants, setting, pattern type, and animal type. Data and code for computational models and statistical analyses are available on OSF: https://osf.io/h2prm/ (DOI 10.17605/OSF.IO/H2PRM).

**3.1.1. Other-reports.** As reported in Table 1, the best fitting model included novelty, learning progress, and expected prediction error in interaction with time and insistence on

**Table 1. Model comparison.** All terms were in interaction with other-reports of insistence on sameness and time. The dependent variable was leave-stay decisions.

| Model | AIC | BIC | Log-Likelihood | Deviance |
|---|---|---|---|---|
| Expected PE, LP, Novelty | 3657 | 3772 | -1812 | 3623 |
| PE, LP, Novelty | 3685 | 3800 | -1825 | 3651 |
| Expected PE, Expected LP, Novelty | 3807 | 3923 | -1887 | 3773 |
| PE, Expected LP, Novelty | 3811 | 3927 | -1889 | 3778 |

sameness. This interaction was significant for learning progress ($\beta$ = -.15, SE = .05, $p$ = .004) and for expected prediction error ($\beta$ = -.10, SE = .05, $p$ = .03). As depicted in Fig 4, participants with lower insistence on sameness scores relied on learning progress to decide whether to disengage at the start of the sampling, but they switched to relying on expected prediction error

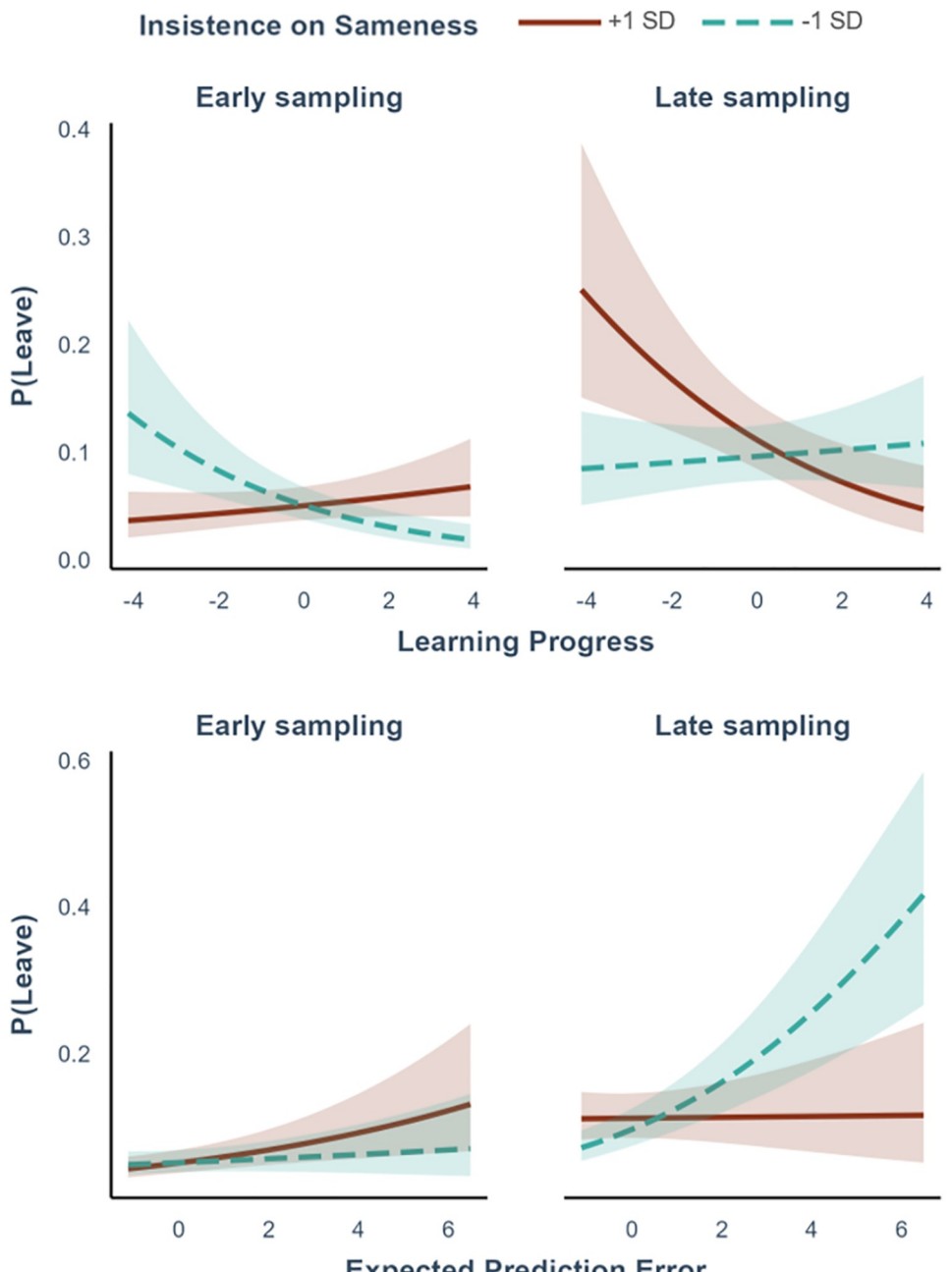

**Fig 4. Results of the logistic regression for leave-stay decisions.** Effects of learning progress and expected prediction error on leave-stay behaviour are shown across time (Early sampling = - 1SD, Late sampling = +1SD) and other-reports of insistence on sameness (low = -1SD, high = +1SD). In the figure, variables are dichotomised to aid interpretability, but they were treated as continuous in the analyses. Participants with lower insistence on sameness scores initially based their decision to disengage on learning progress, but later shifted to expected prediction error. Participants with higher insistence on sameness scores did not rely on either variable initially, but later based their decisions on learning progress. Shaded areas indicate the standard error.

later in the sampling. Participants with higher insistence on sameness did not rely on any of these variables early in the sampling, and they relied on learning progress later in the sampling. The interaction was not significant for novelty (β = .06, SE = .05, *p* = .19). Generating new data from the model parameters led to reproducing the same proportion of leave-stay decisions and the same engagement time observed in the actual data, indicating a good model fit was reached (Fig D in S1 Text).

The full scale for autistic traits showed the same pattern of results, with significant effects for the interactions between learning progress, time, and autistic traits (β = -.16, SE = .06, *p* = .004) and expected prediction error, time, and autistic traits (β = -.10, SE = .04, *p* = .03). Similarly, the subscales for reduced contact and for reduced empathy presented the same pattern of results, while the other subscales did not (see Table A in S1 Text).

**3.1.2. Self-reports.**   When the same model was fit using self-reports, we found a similar trend for expected prediction error in interaction with time and insistence on sameness, which did not reach statistical significance (β = -.0.7, SE = .04, *p* = .077). We found a significant interaction between learning progress and insistence on sameness (β = -.09, SE = .04, *p* = .01) but not between learning progress, insistence on sameness, and time (β = -.03, SE = .04, *p* = .47). When fitting the model on the full scale and the other subscales, the results that were found on the other-reports of insistence on sameness were also found for the reduced contact subscale (see Table B in S1 Text).

## 3.2. The relationship between autistic traits and exploratory decisions

Once participants had decided to stop playing with the animal they were engaged with, they could choose to play with one of the three remaining animals. To determine how they decided what to explore, we ran multinomial logistic models in which the dependent variable consisted of the three available options after each leave decision. The independent variables were expected learning progress, expected prediction error, and the novelty of each option. Multinomial logistic regression does not allow trial-by-trial variables (e.g., learning progress and expected prediction error) to interact with between-subject variables (e.g., ASBQ scores). Hence, to investigate differences across autistic traits, we did a mean split of each trait and ran two independent models for low and high scores.

**3.2.1. Other-reports.**   Results are depicted in Fig 5. When dividing participants according to insistence-on-sameness scores (other-report) with a mean split, we found a significant effect of novelty both for the group with lower insistence on sameness (β = -.85, SE = .22, *p* = <0.001) and high insistence on sameness (β = .63, SE = .20, *p* = .002), but no significant effects of expected prediction error (low group: β = -.15, SE = .14, *p* = .26; high group: β = -.24, SE = .15, *p* = .10) and expected learning progress (low group: β = -.07, SE = .14, *p* = .63; high group: β = .19, SE = .14, *p* = .17). However, it must be noted that for this analysis, the number of participants was low (high group: N = 14; low group: N = 17), and the lack of an effect might be a consequence of the reduced sample size.

**3.2.2. Self-reports.**   When we divided participants on the insistence-on-sameness scores obtained via self-reports, the sample size was bigger (high group: N = 26; low group: N = 31). Here we found a similar effect of novelty for both groups (low group: β = .71, SE = .15, *p* < 0.001, high group: β = .67, SE = .17, *p* < 0.001). Additionally, we found a significant effect of expected learning progress for the high insistence-on-sameness group (β = .30, SE = .11, *p* = .006), but not for the low insistence-on-sameness group (β = -.17, SE = .09, *p* = .06). Conversely, we found a significant effect of expected prediction error for the low insistence-on-sameness group (β = -.20, SE = .10, *p* = .047), but not for the high insistence-on-sameness group (β = -.08, SE = .11, *p* = .46).

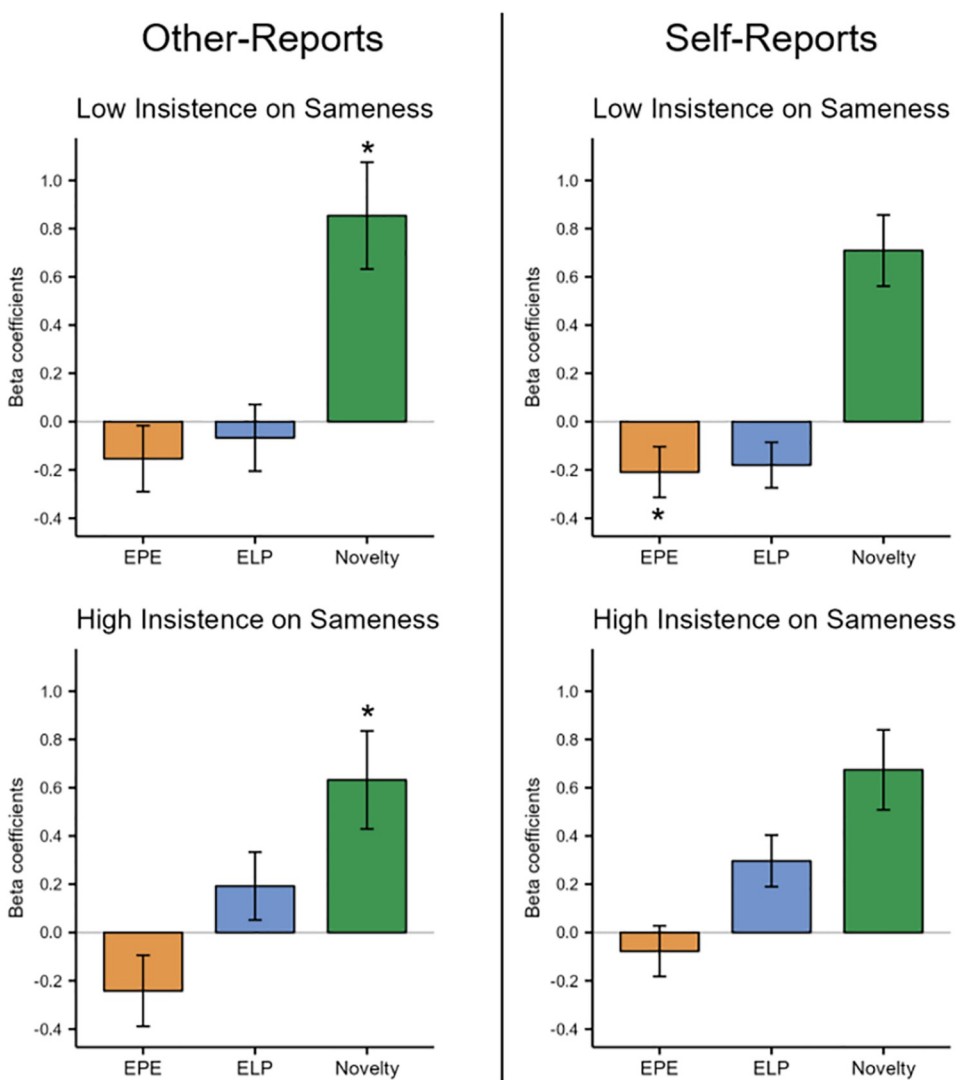

**Fig 5. Results of multinomial logistic regressions for exploratory decisions.** Effects of expected prediction error (EPE), expected learning progress (ELP), and novelty on exploratory choices are shown for high and low insistence-on-sameness traits (mean split) and for other- and self-reports (left and right). All participants picked more novel options. Additionally, when the mean-split was performed on self-reports, participants with low insistence on sameness picked options for which the expected prediction error was low, while participants with high insistence on sameness picked options for which expected learning progress was high. These effects have the same direction in the other-report analyses, but they are not significant. Vertical lines indicate the standard error.

In summary, all participants preferred novel options. In addition, analyses with self-reports indicated that participants with low insistence on sameness scores favored options with lower expected prediction error, while those with high insistence on sameness scores preferred options with higher expected learning progress. Similar effects were observed in the other-report analyses, but they were not statistically significant. Similar multinomial logistic models with the full scale for autistic traits and the remaining subscales, both for self- and other-reports, can be found in the supporting information S1 Text.

### 3.3. Learning performance

We identified differences in exploration drives across the different autistic traits, both in leave-stay decisions and exploratory decisions. We then tested whether these drives were related to differential performance on the learning task. To do this, we tested whether prediction errors decreased over trials for the four hiding pattern types, as a function of autistic traits. Autistic traits were included as a continuous variable. To ease the interpretation of this three-way interaction, we compared a model with the interaction to a model without it using a likelihood ratio test [50]. We found that the interaction was significant ($\chi^2(3) = 10.67$, p = .014). The predictive means (Fig 6) show that higher insistence on sameness scores in other-reports were related to a greater improvement in the performance across all hiding patterns except the high-noise, unlearnable pattern. The interaction was not significant when using self-reports

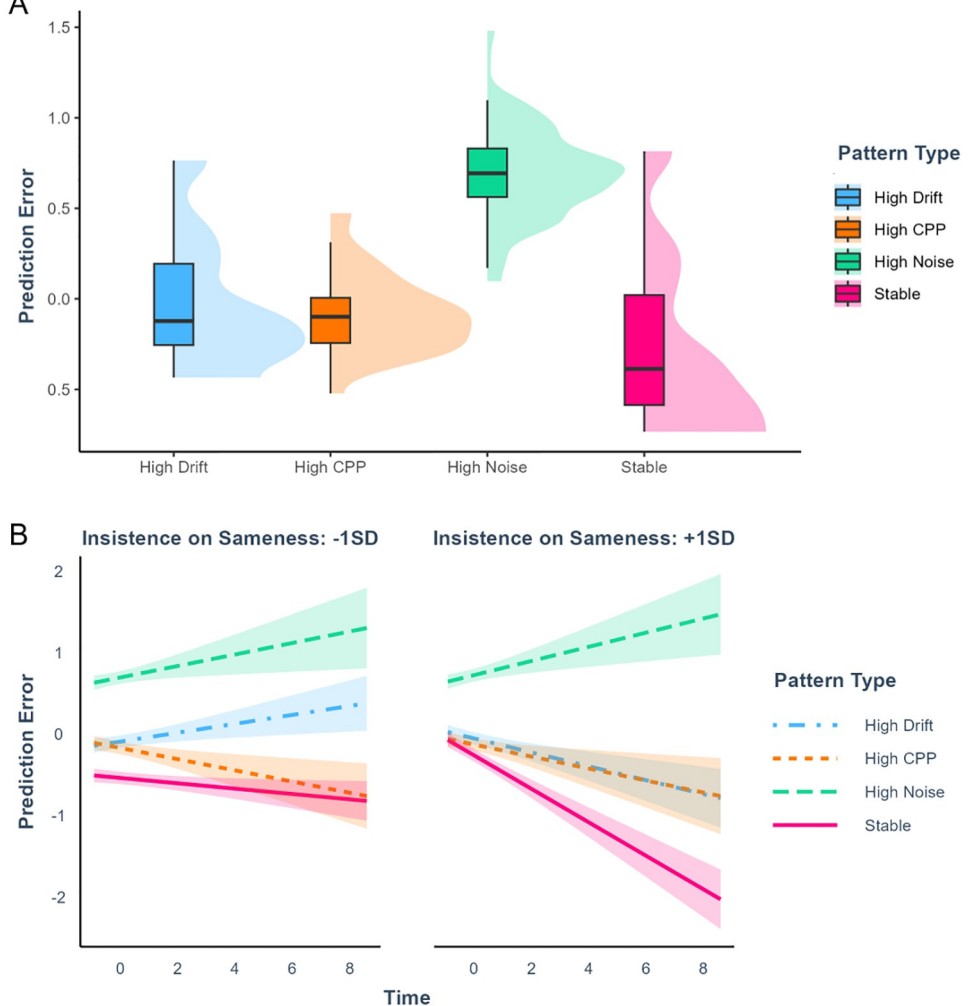

**Fig 6.  A. Distribution of prediction errors over pattern types.** X-axis: pattern types. Y-axis: standardized scores of prediction error. **B. Results for prediction errors over time in relation to autistic traits.** The decrease of prediction errors over time is accentuated when insistence on sameness is higher. Participants' performance on the unlearnable high-noise pattern does not improve. X-axis: Number of consecutive trials played with the same character. Y-axis: standardized scores of prediction error. Autistic traits were treated as a continuous variable, but they are dichotomized in the figure to aid interpretability. Shaded areas indicate the standard error. CPP stands for Change-point probability.

($\chi^2$(3) = 6.32, p = .097). Similar results were also found with the full scale for autistic traits with other-reports ($\chi^2$(3) = 8.53, p = .036), but not with self-reports ($\chi^2$(3) = 4.00, p = .26).

## 4. Discussion

To examine how curiosity-driven exploration relates to autistic traits, we tested participants in a learning task in which they could actively decide when to stop sampling from an environment and what to explore next. Through computational modelling, we obtained trial-by-trial estimates of participants' prediction errors and learning progress, as well as of their *expectations* about future errors and progress. This allowed us to examine which of the factors that have previously been suggested to drive exploration behaviors [4,36,37] determine exploration and learning in people with different degrees of autistic traits. To relate these drives to autistic traits, we collected information about autistic traits in a sample of university students using the ASBQ questionnaire [43], both through other- and self-reports. We found that autistic traits affect curiosity-driven behavior, in terms of leave-stay and exploratory decisions, as well as learning outcomes.

Concerning leave-stay decisions, participants who scored lower on insistence on sameness were more likely to leave an environment if their learning progress was lower during the initial phase of exploration. Hence, they initially relied more on learning progress than participants who scored higher on insistence on sameness. However, during later sampling they stopped relying on learning progress, and exploited expected prediction error instead: They were more likely to leave an activity if they expected to perform badly in the near future. In participants with more insistence on sameness, this pattern of results was different. Participants relied less on learning progress and expected prediction error at the start of the task, and were simply less likely to abandon an environment, showing greater persistence. Only later in time, they started to rely on learning progress, abandoning activities that offered only relatively little learning progress.

When deciding what to explore next, participants preferred more novel options, irrespective of their autistic traits. Yet, participants with lower self-reports of insistence on sameness additionally relied on the expected prediction errors, that is, they preferred options on which they expected to perform better. Conversely, participants with higher self-reports of insistence on sameness relied on the expected learning progress of the available options: They picked the option that offered a greater opportunity to learn. This result can be interpreted in terms of differential utility functions for people with lower and higher insistence on sameness, where the former tried to avoid errors, while the latter tried to maximize learning.

These different drives were related to learning performance. Higher insistence on sameness scores were related to better performance on more probabilistic patterns. This better performance was also observed for the more complex high-drift sequences. Hence, high insistence on sameness is advantageous when tackling complex tasks that require more persistence. Overall, this pattern of results supports our initial hypothesis that insistence on sameness relates to an increased sensitivity to learning progress, which shows in the current task as increased reliance on learning progress to make leave-stay decisions and increased reliance on expected learning progress when deciding what to explore next.

The ability to flexibly adjust to changes in the environments is crucial for effective learning across life [51–53] and it has been found to be reduced in individuals with high autistic traits [26] and in individuals with high attention to detail [27]. This was seen as a form of learning impairment, namely the inability to integrate data to overcome environmental noise [27]. On the contrary, in our task, we observe improvements in performance due to increased insistence on sameness and overall autistic traits. This may be explained by the fact that, in contrast to

previous research, we let participants free to choose the activities they want to learn from, and to learn at their own pace. This might have been especially beneficial for individuals who scored high on insistence on sameness, because their natural inclination to avoid change allowed them to persevere on the task. Conversely, low insistence on sameness might be detrimental in this context, as it might lead individuals to flit from one activity to another without delving deep into any of them.

The pattern of exploratory behaviors was modulated by insistence on sameness in a similar way as by the full scale of autistic traits, as well as some of the subscales, such as reduced contact and reduced empathy for leave-stay decisions, and reduced contact and reduced social interactions for exploratory decisions. These additional effects suggest that the relation between autistic traits and exploration drives is not confined to insistence on sameness, as it extends to a broader range of autistic traits. This result might be due to a causal relation between the different autistic traits. For example, higher insistence on sameness might lead to reduced social interactions [54]. These relations between traits have been found to be stronger in females than males [55], which may explain why we also observe them in our sample, that was in majority composed of female participants. Alternatively, broader genetic [56], chemical [57,58], or brain-related [59,60] factors might impact multiple traits at the same time. Future research should examine the exact brain mechanisms that affect both autistic traits and exploratory behavior, and the causal links between them.

When studying the active aspects of learning, researchers need to find ways of handing control over to the participants without losing experimental rigor. This makes this type of research different from other areas of cognitive science, where paradigms are mostly passive, and participants are required to follow specific instructions (e.g., [61]). Although the active approach comes with unique benefits (i.e., studying aspects of behavior that are driven by intrinsic motivation and engagement), it also requires more data curation. The lack of precise instructions led us to exclude several participants, and more trials were excluded depending on participants' boredom. Future studies should include bigger samples to assess how intrinsically motivated behaviors change as a function of these preprocessing decisions. Finally, when analyzing exploratory decisions only, we were constrained by our analytic tools to split the sample in two groups, and look at each subgroup's results independently. This resulted in a loss of granularity in the analysis of exploratory decisions compared to our analysis of leave-stay decisions. Future studies are needed to develop analytic methods to examine traits as continuous even in multinomial models.

## 5. Conclusions

In this study, we provide insights into how autistic traits influence exploration behavior. By using a novel exploration task and a hierarchical delta-rule model, we show that individuals with lower insistence-on-sameness scores are more likely to rely on learning progress early on but switch to expected prediction error later. Instead, individuals with higher scores on the same subscale are less likely to abandon an activity, showing greater persistence, and eventually start to rely on learning progress. We also show that individuals with lower insistence on sameness are more likely to choose options with low expected prediction error, while those with higher insistence on sameness tend to select options with high expected learning progress. These findings provide a nuanced understanding of how autistic traits influence exploration behavior and learning outcomes. Shifting the focus from what individuals can do to what they are motivated to do, we show how learning abilities change across different degrees of autistic traits.

## Supporting information

**S1 Text. Supplementary tables, figures, and analyses.** Fig A. The preprocessing pipeline. Fig B. Participants' predictions. Fig C. Model simulations. Fig D. Parameters recovery. Fig E. Model comparison. Fig F. Comparing behavioral data to the predictions of the logistic model. Table A. Relation between leave-stay decisions and other-reports. Table B. Relation between leave-stay decisions and self-reports. Appendix A. Task-related questionnaire. Appendix B. Supplementary analyses.
(DOCX)

## Author Contributions

**Conceptualization:** Francesco Poli, Marlene Meyer, Rogier B. Mars, Nanda Rommelse, Sabine Hunnius.

**Data curation:** Maran Koolen.

**Formal analysis:** Francesco Poli, Carlos A. Velázquez-Vargas, Jessica Ramos-Sanchez.

**Funding acquisition:** Rogier B. Mars, Sabine Hunnius.

**Investigation:** Maran Koolen.

**Methodology:** Francesco Poli.

**Software:** Francesco Poli, Maran Koolen.

**Supervision:** Marlene Meyer, Rogier B. Mars, Nanda Rommelse, Sabine Hunnius.

**Visualization:** Francesco Poli, Carlos A. Velázquez-Vargas.

**Writing – original draft:** Francesco Poli.

**Writing – review & editing:** Maran Koolen, Carlos A. Velázquez-Vargas, Jessica Ramos-Sanchez, Marlene Meyer, Rogier B. Mars, Nanda Rommelse, Sabine Hunnius.

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
