## [Decision Letter · Decision Letter 0]

15 Nov 2023

Dear Mr Poli,

Thank you very much for submitting your manuscript "Autistic traits foster effective curiosity-driven exploration" for consideration at PLOS Computational Biology.

As with all papers reviewed by the journal, your manuscript was reviewed by members of the editorial board and by several independent reviewers. In light of the reviews (below this email), we would like to invite the resubmission of a significantly-revised version that takes into account the reviewers' comments.

**Specifically, all reviewers raise important concerns about fitting procedure, missing details and descriptions of basic performance of the task, lack of relationship between estimated model parameters and self-report measures, all of which should be carefully addressed in the revised manuscript.**

We cannot make any decision about publication until we have seen the revised manuscript and your response to the reviewers' comments. Your revised manuscript is also likely to be sent to reviewers for further evaluation.

Sincerely,

Alireza Soltani

Academic Editor

PLOS Computational Biology

Zhaolei Zhang

Section Editor

PLOS Computational Biology

Reviewer's Responses to Questions

**Comments to the Authors:**

Reviewer #1: The review is uploaded as an attachment

Reviewer #2: The authors report on whether individual differences in behavior in a patch foraging task can be related to autistic self-report measures. The task is cleverly designed, and the modeling of the behavior is quite extensive. However, the overall value of the computational modeling of behavior is lessened by many unexplained and confounding choices in the data analytic strategy and the authors' inability to ground their modeling in basic behaviors. Overall, the study seems underpowered to reliably assess how individual differences in autistic traits correlate with curiosity-driven exploration.

Please provide more details in terms of the percentage of trials thrown away for the included participants based on the boredom exclusion criteria. Also, the authors use logistic models to predict when participants abandoned choosing a particular animal and switched to another animal; these rely on trial-by-trial estimates. Were the trials overlapping with boredom not included in fitting the hierarchical reinforcement learning model, or were they simply censored? Overall, there appears to be a great deal of data curation, which is concerning in relation to the relatively small effect sizes reported when the task behavior is related to the autism-related self-reports.

Another important caveat is that the authors only found a relationship between the model-derived reinforcement learning (RL) estimates and the other-reports. Substantially weaker or null effects were found when the same RL measures were correlated with self-reports. This is concerning because the authors report that only 60% (42 out of the 70 participants) handed in the other self-report questionnaire, and therefore the sample sizes are very different.

The novelty of each option utilizes kappa as a fixed parameter, but the value of this parameter and how it was determined is not explained. This is important since the best-fitting model includes trial-by-trial novelty estimates.

There is no basic presentation of the participants' behaviors within the task and how they scale with individual difference measures. We must rely entirely on the authors' statement that the behavior was well fit without any direct evidence. I would like to see some basic behaviors plotted in relation to key decision criteria in the task, such as the frequency of switching between animals, the duration of sampling within a patch (i.e., animal), etc., and then how well the model can predict those behaviors (independent of the individual differences). Otherwise, it is very difficult to judge the validity of the modeling in relation to the individual differences.

Reviewer #3: PLos Review

I thank the authors for submitting this paper, which was read with great interest. Poli et al., make use of a free choice exploration task in which participants choose to learn about the hidden location of a set of four animals across different contexts, whose location were determined by a distinct (learnable) path. The authors have done a good job of bringing attention to the understanding of autistic traits (insistence on sameness) on learning and free choice exploration and I agree with the use of modelling approaches to understand the computational signatures of these traits, and the emphasis on understanding individual differences in these traits as opposed to diagnosis. However, there are some issues that limit my enthusiasm for this manuscript, which should be be addressed before the manuscript would be considered suitable for publication.

Major points

1. Computational modelling: Given that in its current form, results of the manuscript hinge on the output of the computational modelling, I want to draw attention to a number of issues that we had with the modelling itself:

a. My primary concern is that whilst the model has been used in previous work from the same group, we would ask the authors to include some model quality check given current guidelines to aid in the reliability, interpretation and identifiability of parameters (see Wilson & Collins 2019 eLife, Palminteri et al 2017 TICS). For example, starting with simulating some data (as undertaken in previous work) to see if the key behavioural features of the task can be recapitulated (see point d. below), but also demonstrating the recoverability of the parameters. It is also conceivable that despite the sophistication of the model used in the paper, a simpler model may be able to explain the data similarly well. As such, I would recommend the inclusion of model comparisons.

b. I struggled to understand some aspects of the modelling, and think that the description of the modelling details and approach could be clearer. For example, it is not immediately clear how many parameters are being fitted, which are participant specific or which, if any, are specific to the four different animals (contexts) introduced in the task.

c. To help the above point, the authors could draw more explicit links between the model details in the Methods section and Figure 3. The legend for Figure 3 could offer more explanation of the expressions on the right hand side of the figure

d. Finally, whilst the paper clearly has a computational focus, reporting and plotting some model agnostic results may help the readers understanding. For instance, a demonstration of the effects of the different location trajectories (e.g. high drift, high changepoint probability etc) on p(leave) may be useful. It would also be reassuring to see individual datapoints for the key model agnostic results plotted.

2. Trait measures: In its current form, the manuscript focuses on individual differences, which I consider an important step in understanding behaviour. However, I would like a few points to be clarified, or the focus of the paper to be adjusted to accommodate some concerns

a. Most of the significant results in the manuscript rest on “other” (parent) reports of autistic traits, rather than people’s self-report. Firstly, this is not clear from the abstract or title, and seems to be an important omission.. I am not aware of any literature validating or advocating for the use of other people’s reports of autistic traits in adults, perhaps the authors can cite some papers supporting the reliability of this approach? Additionally, could the authors provide some motivation behind asking a parental figure to complete the ‘other’ reports, given the participants in this study are all adults. How should the reader interpret the findings where ‘other’ reported autistic traits do not align with the findings of ‘self’ reported autistic traits? Was there a specific hypothesis about other vs self-report measures? If so a direct statistical comparison of these measures should be included.

b. We would ask the authors to report more information about the ASBQ scores in this sample, for instance mean scores and ranges for the total score and subscales. It would also be helpful to see the relationship between self-report and other-report scores within participants.

c. The authors rightly seek to take the focus away from diagnosis and avoid groups, but then in the results binarize autistic traits into high or low. Aside from the problem of dichotomizing continuous variables (see Altman & Royston 2006 BMJ), could the authors explain why they chose to dichotomize (a movement back towards groups) in order to understand individual differences?

3. Framing of the paper: In its current form, the manuscript has many interesting threads, but readability and understanding may be improved if the number of threads was reduced

a. For instance, upon reading the whole paper, it is not clear how curiosity fits into the story with insistence on sameness? Could the authors either make this clearer, or make this thread slightly less apparent throughout the manuscript?

b. Additionally, in the methods, a factor of location trajectory (i.e. high drift, high noise etc) is proposed, but this does not really feature in the first two sections of the results, and is only mentioned briefly in the discussion. It may be the case that the output of the computational modelling includes this factor, but could the authors please clarify or expand on the role of this factor.

We also have the following minor notes:

1. We note a typo after the equation in Section 3.1 “where the autistic..”, where should be omitted.

2. Greater explanation of the error bars in Figure 4 and 5 would be helpful, given the complexity of interpreting high level interactions

3. It is not fully clear what constitutes a trial – perhaps Figure 1 could be reworked to indicate a full trial sequence in order along a row.

4. The authors refer to the model used as a hierarchical reinforcement learning model, which we think is a slight misuse of reinforcement, as there are no rewards in the task. We think a hierarchical learning model/hierarchical delta rule model may be more appropriate.

**Have the authors made all data and (if applicable) computational code underlying the findings in their manuscript fully available?**

Reviewer #1: Yes

Reviewer #2: Yes

Reviewer #3: Yes

PLOS authors have the option to publish the peer review history of their article (what does this mean?). If published, this will include your full peer review and any attached files.

Reviewer #1: **Yes: **Jae Hyung Woo

Reviewer #2: No

Reviewer #3: No
---

## [Decision Letter · Decision Letter 1]

29 Jun 2024

Dear Mr Poli,

Thank you very much for submitting your manuscript "Autistic traits foster effective curiosity-driven exploration" for consideration at PLOS Computational Biology. As with all papers reviewed by the journal, your manuscript was reviewed by members of the editorial board and by several independent reviewers. The reviewers appreciated the attention to an important topic. Based on the reviews, we are likely to accept this manuscript for publication, providing that you modify the manuscript according to the review recommendations.

**Specifically, in addition to addressing the final comments from Reviewer #1, please provide a more detailed response to point 4 of Reviewer #2, which is similar to point 1d from Reviewer #3. Consider including a new figure that compares the average performance of all participants with average model estimates, which could complement what is shown in Figure 2B. Additionally, incorporating some model-agnostic results would enhance the manuscript for readers. Finally, please include a Data Availability section to your manuscript. Please note that your revised manuscript will not undergo further external review but will be evaluated at the editorial level.  **

Sincerely,

Alireza Soltani

Academic Editor

PLOS Computational Biology

Zhaolei Zhang

Section Editor

PLOS Computational Biology

Reviewer's Responses to Questions

**Comments to the Authors:**

Reviewer #1: I appreciate the authors’ effort in addressing all the points raised in the review. I just have one follow-up to the first major comment as below:

Comment #1:

Thank you for running this additional model fitting. It is now more clear which component of the hierarchical model becomes more relevant depending on the type of environment.

Yet, I’d still like to get some more clarifications on why this particular group-level fitting scheme was adopted, especially regarding whether group-fitting is appropriate for inferring individual variability. (This point was not addressed in the response letter.) I see that the group-level fit (dic_a or dic_ab) is overall better than the individual-level fit (dic_ai or dic_abi), but isn’t this again a comparison at a group level by averaging across all subjects? More precisely, aren’t the DIC of dic_ai and dic_abi models in Figure S5 computed by taking average across individuals? What I am trying to get at is whether there is any variability in terms of which model best account for each participant’s behavior (dic_ai vs. dic_abi).

Also, Figure 3 caption states “All parameters were estimated across all individuals, but separately for each animal a,” but the following methods section (below Eq. 8) mentions only the learning rates. These two can be made consistent.

Comment #2.

Thank you for adding the additional panel for the example session. It is now much more clear and intuitive.

Comment #3:

Thank you for running this extra analysis on RT. The results are interesting but I agree that it is better kept out of the manuscript.

Typo - caption for Figure S2: every “patter” of every environment

Reviewer #2: The authors have addressed my prior concerns.

Reviewer #3: The authors have for the most part addressed my concerns satisfactorily and I am now happy to recommend publication.

**Have the authors made all data and (if applicable) computational code underlying the findings in their manuscript fully available?**

Reviewer #1: Yes

Reviewer #2: Yes

Reviewer #3: Yes

PLOS authors have the option to publish the peer review history of their article (what does this mean?). If published, this will include your full peer review and any attached files.

Reviewer #1: No

Reviewer #2: No

Reviewer #3: No

Figure Files:

Data Requirements:

Reproducibility:

References:

---

## [Editor Report · Decision Letter 2]

3 Sep 2024

Dear Mr Poli,

We are pleased to inform you that your manuscript 'Autistic traits foster effective curiosity-driven exploration' has been provisionally accepted for publication in PLOS Computational Biology.

Best regards,

Alireza Soltani

Academic Editor

PLOS Computational Biology

Zhaolei Zhang

Section Editor

PLOS Computational Biology

---

## [Editor Report · Acceptance letter]

2 Oct 2024

PCOMPBIOL-D-23-01580R2 

Autistic traits foster effective curiosity-driven exploration

Dear Dr Poli,

I am pleased to inform you that your manuscript has been formally accepted for publication in PLOS Computational Biology. Your manuscript is now with our production department and you will be notified of the publication date in due course.

With kind regards,

Lilla Horvath
